# Simulation of Cardiac Flow under the Septal Defect Based on Lattice Boltzmann Method

**DOI:** 10.3390/e24020187

**Published:** 2022-01-27

**Authors:** Zhengdao Wang, Xiandong Zhang, Yumeng Li, Hui Yang, Haihong Xue, Yikun Wei, Yuehong Qian

**Affiliations:** 1State-Province Joint Engineering Lab of Fluid Transmission System Technology, Faculty of Mechanical Engineering and Automation, Zhejiang Sci-Tech University, Hangzhou 310018, China; dao@zstu.edu.cn (Z.W.); 201930504021@mails.zstu.edu.cn (X.Z.); 201920501036@mails.zstu.edu.cn (Y.L.); yanghui@zstu.edu.cn (H.Y.); 2Department of Pediatric, Xinhua Hospital, School of Medicine, Shanghai Jiao Tong University, Shanghai 200092, China; 3School of Mathematical Science, Soochow University, Suzhou 215006, China; yuehongqian@suda.edu.cn

**Keywords:** complex boundary, cardiac flow, entropy generation rate, lattice Boltzmann method

## Abstract

In this paper, the lattice Boltzmann method was used to simulate the cardiac flow in children with aseptal defect. The inner wall model of the heart was reconstructed from 210 computed tomography scans. By simulating and comparing the cardiac flow field, the pressure field, the blood oxygen content, and the distribution of entropy generation before and after an operation, the effects of septal defect on pulmonary hypertension(PH), cyanosis, and heart load were analyzed in detail. It is found that the atrial septal defect(ASD) of the child we analyzed had a great influence on the blood oxygen content in the pulmonary artery, which leads to lower efficiency of oxygen binding in the lungs and increases the burden on the heart. At the same time, it also significantly enhanced the entropy generation rate of the cardiac flow, which also leads to a higher heart load. However, the main cause of PH is not ASD, but ventricular septal defect (VSD). Meanwhile, it significantly reduced the blood oxygen content in the brachiocephalic trunk, but rarely affects the blood oxygen contents in the downstream left common carotid artery, left subclavian artery, and descending aorta are not significantly affected by VSD. It causes severe cyanosis on the face and lips.

## 1. Introduction

Congenital heart disease (CHD), having a high incidence (about 0.8–1.2% of newborns) in China and the United States, becomes one of the main diseases affecting the survival rate of newborns [1,2]. As one of the main causes of CHD, the septal defect can be divided into the atrial septal defect (ASD) and ventricular septal defect (VSD) according to the location of the septal defect [3,4]. The septal defect causes abnormal changes in hemodynamics, precordial pain, and fatigue of the patient. The severe patients are prone to growth retardation, pale complexion, shortness of breath, cyanosis of the skin, and even premature death [5]. In order to analyze the cardiac flow and to understand the impact of the septal defect, the flow process was visualized by experiments and numerical simulations [6,7,8].

Hemodynamic data can be obtained from imaging examinations, such as cardiac catheterization, echocardiography, computed tomography (CT), magnetic resonance imaging(MRI), etc. However, all these examinations have certain limitations. Cardiac catheterization can accurately measure the pulmonary artery pressure and hemodynamic data. However, it is an invasive examination [9]. Serious complications may occur during pre- and post-operation. In addition, it is expensive and with radiation. Echocardiography provides non-invasive imaging data in pediatric cardiology and with a low cost of inspection [10]. Echocardiography can visualize complicated flow with a heart chamber or valve motion, however, with limitations in temporal and spatial resolutions. A cardiac CT scan is an important low-invasive diagnostic tool for the evaluation of congenital heart disease in children. The main drawback of CT is the cumulative radiation dose from repeat examinations pre- and post-operation or endovascular treatment [11]. Cardiac MRI is superior to CT and echocardiography, as well as hemodynamic assessment by cardiac catheterization. The main limitation of MRI is the relatively long scan duration for the required spatiotemporal resolution [12]. With the rapid development of computer and medical image processing technology, computational fluid dynamics (CFD) has become an important tool for studying blood flow dynamics. CFD is noninvasive. It can provide quantitative analysis and qualitative visualization of many crucial hemodynamic parameters such as velocity, pressure, and blood oxygen content in hemodynamic studies [13]. CFD simulations of cardiac flow in the whole heart with ASD or VSD have rarely been reported due to the complex geometric boundary. The complex geometric properties of the cardiovascular system and the complex circulations also lead to the problems such as the reconstruction of the 3D cardiac model [1,14], large deformation of the heart, and the flow in the systemic/pulmonary circulations. The solution to these problems is of great significance for the simulation of cardiac flow and the diagnosis of disease.

With the development of computer technology, computational fluid dynamics (CFD) is becoming more and more mature. It provides reliable numerical results for the study of cardiac flow [15,16,17]. Over the past 30 years, the lattice Boltzmann method (LBM) has become one of the most popular methods in CFD. It has great parallelism and has a strong advantage in dealing with complex boundaries [18,19,20,21,22]. The traditional grid method is difficult to deal with the complex inner wall of the heart. Especially when dealing with the large deformation of hearts, the quality of the grid is extremely low. Because the Cartesian grid is used in LBM, it has an excellent advantage in dealing with the cardiac flow with complex boundaries. Due to the difficulty in simulating the blood oxygen content by the traditional method, the simulation of blood oxygen content is rarely seen in existing studies. However, the distribution of blood oxygen content is crucial to explain and understand the cyanosis of children with CHD. In this work, the DDF (Double Distribution Function) approach in LBM proposed by Guo [18,19] is used to simulate the cardiac flow with the inner wall of the heart as its flow boundary. Two distribution functions are used to simulate the flow field and the blood oxygen content, respectively. The visualization of the simulation can help us better understand the mechanical performance of the heart and the distribution of blood oxygen content in the heart.

The viscous entropy generation was introduced to understand the energy dissipation in heat flow and the performance of the septal defect on the heart burden of children. The expression of the viscous entropy is defined as [22,23]:
(1)$${\dot{S}}_{u}=\frac{\mu }{T}\left\{2\left[{\left(\frac{\partial u}{\partial x}\right)}^{2}+{\left(\frac{\partial v}{\partial y}\right)}^{2}+{\left(\frac{\partial w}{\partial z}\right)}^{2}\right]+{\left(\frac{\partial u}{\partial y}+\frac{\partial v}{\partial x}\right)}^{2}+{\left(\frac{\partial u}{\partial z}+\frac{\partial w}{\partial x}\right)}^{2}+{\left(\frac{\partial v}{\partial z}+\frac{\partial w}{\partial y}\right)}^{2}\right\}$$
where *μ* is the dynamic viscosity of blood and *T* is the body temperature. *u*, *v*, and *w* represent the *x*-, *y*- and *z*-component of velocity, respectively.

## 2. Modeling

We selected a child with both ASD and VSD as the object to explore the effects of ASD and VSD on blood flow in the heart. The child’s heart is reconstructed based on the CT scans. The heart structure after an operation is simulated by filling the septal defects. The geometry of the heart was extracted from 210 CT scans provided by Dr. Xue, who is working in Xinhua hospital in Shanghai, China. The ethics approval is included in the attachment. The following model was applied to extract the inner wall boundary of the heart.
(2)$$\frac{\partial R}{\partial t}=\max\left\{0,C_{1}\left(I-1\right)+1\right\}\cdot \nabla^{2}R$$
where *C*_1_ is the value relevant to the grayscale of the extracted object in the CT images. *I* is the grayscale of the image with the range of 0–1. *R* represents the dimensionless characteristic of the extracted object with the range of 0–1. To accelerate the image extraction, the value of *R* > *C*_2_ is set to 1 after each iteration. In this research, *C*_2_ is chosen as 0.7. At the end of the extraction, the inner wall boundary of the heart is obtained by the 0.5 contours of *R*. According to the scale, the extracted inner wall boundaries are integrated into a complete heart after interpolation and filtering. The whole reconstruction process of the 3D cardiac model is shown in Figure 1.

Figure 2 shows the final complete cardiac model. It is worth noting that the extracted model uses the inner wall of the heart which is the boundary of actual cardiac flow. Because the complex structure of the inner wall is a great challenge for the grid, the smooth outer wall was often used as the flow boundary in previous numerical studies [24]. Due to the advantages of LBM in complex boundary treatment, the inner wall of the heart is used for better describing the actual cardiac flow.

AA (aortic arch), SVC (superior vena cava), PA (pulmonary artery), LA (left atrium), RA (right atrium), LV (left ventricle), and RV (right ventricle) are marked in Figure 2.

## 3. Numerical Algorithm and Boundary Conditions

The governing Equations of cardiac blood flow are the Navier-Stokes Equations. In order to analyze the effect of the septal defect on cardiac flow, the convective-diffusion equation is used to express the dimensionless oxygen content [8,15].
(3)$$\frac{\partial \rho }{\partial t}+\nabla \cdot \left(\rho u\right)=0$$
(4)$$\frac{\partial \left(\rho u\right)}{\partial t}+\nabla \cdot \left(\rho uu\right)=-\nabla p+2\mu \nabla \cdot S$$
(5)$$\frac{\partial C_{b}}{\partial t}+u\cdot \nabla C_{b}=\alpha \nabla^{2}C_{b}$$
where *ρ* is the blood density; ***u*** the velocity of blood flow; *p* the dimensionless local pressure of blood (0 < *p* < 1; 0: the diastolic pressure; 1: systolic pressure); ***S*** the strain rate tensor of blood; *μ* the viscosity of blood; *α* the diffusion coefficient of blood oxygen content; *C_b_* the dimensionless blood oxygen content (−1 < *C_b_* < 1; −1: the oxygen content in the superior/inferior vena cava; 1: the oxygen content in the pulmonary vein).

The evolution equation of the DDF approach in LBM is expressed as follows [19].
(6)$$f_{i}\left(x+c_{i}\delta t,t+\delta t\right)=f_{i}\left(x,t\right)-\omega_{f}\left[f_{i}\left(x,t\right)-f_{i}^{\left(eq\right)}\left(x,t\right)\right]$$
(7)$$g_{i}\left(x+c_{i}\delta t,t+\delta t\right)=g_{i}\left(x,t\right)-\omega_{g}\left[g_{i}\left(x,t\right)-g_{i}^{\left(eq\right)}\left(x,t\right)\right]$$
where *f_i_* distribution simulates the flow field; *g_i_* simulates the blood oxygen content distribution. Different from the previous method [19], we eliminated the external force term in Equation (6). The additional external force term is the driving force in convection. It is not the main cause of cardiac flow. To build the relationship between the mesoscopic method and the macroscopic variables, the following statistical relationship is used: $\rho =\sum_{i}f_{i}\left(x,t\right)$; $\rho u=\sum_{i}\left[f_{i}\left(x,t\right)c_{i}\right]$; $C_{b}=\frac{1}{\rho }\sum_{i}g_{i}\left(x,t\right)$. In this paper, we used the classic D3Q19 model [18]. The schematic of the discrete velocities is shown in Figure 3.

According to Ref. [18], the discrete velocities and the weight coefficients are chosen as follows (Table 1).

The equilibrium distribution functions *f_i_*^(*eq*)^ and *g_i_*^(*eq*)^ in Equations (6) and (7) have second-order Maxwell distribution form [18]:(8)$$f_{i}^{\left(eq\right)}=\rho w_{i}\left[1+\frac{c_{i}\cdot u}{c_{s}^{2}}+\frac{{\left(c_{i}\cdot u\right)}^{2}}{2c_{s}^{4}}-\frac{u^{2}}{2c_{s}^{2}}\right]$$
(9)$$g_{i}^{\left(eq\right)}=\rho C_{b}w_{i}\left[1+\frac{c_{i}\cdot u}{c_{s}^{2}}+\frac{{\left(c_{i}\cdot u\right)}^{2}}{2c_{s}^{4}}-\frac{u^{2}}{2c_{s}^{2}}\right]$$

Using the Chapman-Enskog technique, one can obtain the macroscopic governing equations Equations (3)–(5) from the mesoscopic equations, Equations (6) and (7) for incompressible flow. Meanwhile, the relation between the relaxation factor *ω_f_* and the dynamic viscosity *μ*, and the relation between the relaxation factor *ω_g_* and the diffusion coefficient of blood oxygen content *α* can be obtained by the Chapman-Enskog technique [19]:(10)$$\mu =\rho c_{s}^{2}\left(\frac{1}{\omega_{f}}-\frac{1}{2}\right)\delta t$$
(11)$$\alpha =c_{s}^{2}\left(\frac{1}{\omega_{g}}-\frac{1}{2}\right)\delta t$$

The boundary condition of cardiac flow is a non-slip boundary at the inner wall of the heart. For blood oxygen content, the adiabatic boundary condition in thermodynamics is used to implement the non-impermeability boundary conditions of blood oxygen. In LBM, the above two boundary conditions are expressed as follows [22,25].
(12)$$f_{\bar{i}}\left(x,t+\delta t\right)=f_{i}^{+}\left(x,t\right)$$
(13)$$g_{i^{\prime }}\left(x+c_{i^{*}}\delta t,t+\delta t\right)=g_{i}^{+}\left(x,t\right)$$
where $\bar{i}$ is the direction opposite to i; $i^{\prime }$ the direction symmetrical to i about the wall; $c_{i^{*}}=\left(c_{i}+c_{i^{\prime }}\right)/2$ the tangential component of discrete velocity ***c****_i_* along the wall surface. The dynamics boundary condition of the inflows and the outflows are free-flow boundaries. The blood oxygen content of inflows from the superior/inferior vena cavas is set as low blood oxygen content (*C_b_* = −1). The blood oxygen content of inflows from the pulmonary vein is set as high blood oxygen content (*C_b_* = 1), respectively. The blood oxygen content of outflows are free boundary conditions.

## 4. Results and Discussion

### 4.1. Analysis of Cardiac Flow of the Preoperative Child

Pressure in LA and LV is relatively high. Before the operation, the fluid flows from LA and LV with high blood oxygen content into RA and RV through ASD and VSD, respectively. Figure 4 shows the simulation results of blood flow and blood oxygen content distribution in the heart of the child patient.

In Figure 4, the colors are used to mark the dimensionless blood oxygen content where the streamline passes through. Red refers to relatively high blood oxygen content (*C_b_* = 1); blue to relatively low blood oxygen content (*C_b_* = −1). Before the operation, the blood oxygen content in PA is much higher than that in vena cava. By comparing the flow fields and blood oxygen content distributions near ASD and VSD, the main reason for the decrement of the blood oxygen content in the PA is the mixing flow at ASD. A distinct jet flow from LA to RA is formed at ASD. This jet carries blood with relatively high blood oxygen content in LA into RA and rapidly increases the blood oxygen content in RA. As the child had severe VSD, there is no obvious jet flow at the VSD. It proves that the main cause of pulmonary hypertension (PH) is this severe VSD. The special VSD structure cut apart blood from LV. A tiny fraction of relatively low oxygen content blood (*C_b_* ≈ 0.3) then streams into ascending aorta. We zoom in on the flow fields and blood oxygen content distributions at AA and PA downstream of blood flow on both sides in Figure 5 to further analyze the effect of septal defects.

Figure 5 shows that the blood with low blood oxygen content on the anterior side of AA mainly enters BT (Brachiocephalic Trunk). Therefore, VSD reduces the blood oxygen content in BT. The non-dimensional blood oxygen content in BT has a minimum lower than 0.34 and an average of 0.64. It decreases by about 23% of the difference in arterio and venous oxygen content. It causes severe cyanosis on the face and lips and is consistent with the clinical reports. The blood oxygen contents in the downstream LCCA (left common carotid artery), LSA (left subclavian artery), and DA (descending aorta) are not significantly affected by the septal defect. The blood in PA, downstream of RA and RV, has low blood oxygen content. With the dimensionless blood oxygen content of about −0.5, it increases by about 25% of the difference in arterio and venous oxygen content.

Figure 6 shows the flow field, the pressure field, the blood oxygen content distribution, and the entropy production on the horizontal section of ASD and VSD to analyze the effect of the septal defect on blood flow in the heart.

Figure 6a shows a strong jet from LA to RA at ASD. Besides the strong mixing of blood oxygen content produced by the jet, it also has large velocity entropy according to the result of entropy generation, thus indicating strong energy dissipation. However, because of the narrow passage between LA and RA. ASD didn’t cause severe PH. The VSD is more severe than the ASD for this child. From Figure 6b, it is seen that the pressure in RV reaches a high level. However, due to the low-pressure difference between LV and RV, no obvious jet flow from LV to RV is formed. It leads to low blood oxygen content mixing and no obvious rise of entropy generation at VSD. It is also found that the entropy generation rate is extremely high at ASD.

### 4.2. Comparison of Cardiac Flows of the Child before and after the Operation

The septal defect is filled after the operation. Figure 7 shows the simulated streamline and blood oxygen content results.

As shown in Figure 7, the blood with relatively high oxygen content in LA and LV is no longer mixed with the blood with low oxygen content in RA and RV due to the filling of the septal defect after the operation. Therefore, the blood oxygen content returns to normal levels in AA and PA. The cure of ASD was eliminated in the jet flow from LA to RA. It stabilized the cardiac flow in LA and reduces energy loss. The cure of VSD makes high blood pressure in LV no longer affecting the blood pressure in RV. It allows PH to be alleviated. The distributions of flow entropy in the heart before and after the operation are compared to analyze the effect of the operation on the blood flow dissipation in the heart (see Figure 8).

The energy generation rate can value the energy loss of flow. Higher energy loss leads to a higher heart load. The large bend of SVC (Superior Vena Cava) and three branches above AA have high flow entropy generation. Besides, the entropy generation at the tricuspid and mitral valves of the heart is also high. The abnormal high point of entropy production at ASD is eliminated after the operation. Figure 8 shows that the cure of ASD eliminates the entropy generation in the interval above 10^−11^. Figure 9 shows the results of probability density function *f*(*x*) × d*x* of entropy generation and flow velocity before and after operation for comparison. The entropy generation in the heart is mainly between 10^−15^ and 10^−10^. Meanwhile, Figure 9a shows that the entropy generation varying from 10^−14^ to 10^−12^ is also eliminated after the operation. It is believed that the operation can eliminate both high and medium entropy generation rates.

Figure 9b shows that the low-speed flow decreases, and the high-speed flow increases after the operation. The structure of cardiac blood flow after the operation is simpler than that before the operation. The low-speed complex flow structure is eliminated, the flow becomes more concentrated. It also proves that the energy dissipation of cardiac flow decreases and cardiac load is reduced after the operation.

## 5. Conclusions and Future Work

The work simulated the cardiac blood flow of a child with ASD and VSD. The DDF approach of LBM was implemented to stimulate the velocity field, the pressure field, and the oxygen content distribution in the heart before and after the operation. The main conclusions are as follows.

A. In this case, a strong jet for LA to RA is formed at ASD due to the pressure difference between LA and RA. The pressure in RA is not significantly affected by ASD. Meanwhile, there was no jet flow at VSD. The pressure in RV is significantly raised because of the VSD. Thus the main cause of PH is the severe VSD in this case.

B. Because the jet at ASD carried the blood with high oxygen content in LA to RA., the blood oxygen content in PA increases by about 25% of the difference in arterio and venous oxygen content. It reduces the efficiency of oxygen binding in the lungs and increases the burden on the heart. Meanwhile, the flow at VSD is approximately laminar flow. It reduced the blood oxygen content in BT by about 23% of the difference in arterio and venous oxygen content. It is believed to be the main cause of cyanosis on the face and lips. The blood oxygen contents in downstream LCCA, LSA, and DA were not significantly affected by VSD.

C. The septal defect exacerbated the energy loss of cardiac flow, which means high heart load. In this case, the energy loss, which is represented by a high entropy generation rate, is mainly caused by ASD. Besides, the complex flow caused by the septal defect significantly increased the medium entropy generation rate.

D. After the operation, the blood oxygen content in BT and PA returned to a normal level because the septal defect was filled. The PH is eliminated. The energy loss is also reduced which means the decrement of cardiac load.

The fixed boundary conditions are not accurate enough to describe the flow simulation in the heart. The periodic flow is more accurate and can achieve more details of the cardiac flow compared with present quasi-steady results. In future work, the large deformation model with moving boundaries will be simulated by IB-LBM (Immersion Boundary-Lattice Boltzmann Method). Meanwhile, free flow boundary conditions cannot guarantee that the mass flow at the inlet and outlet of circulations is equal. Simplified circulation models are also necessary for future work to achieve a more accurate prediction of blood flow volume and the local pressure, and to provide better analysis of CHD.

## Figures and Tables

**Figure 1 entropy-24-00187-f001:**
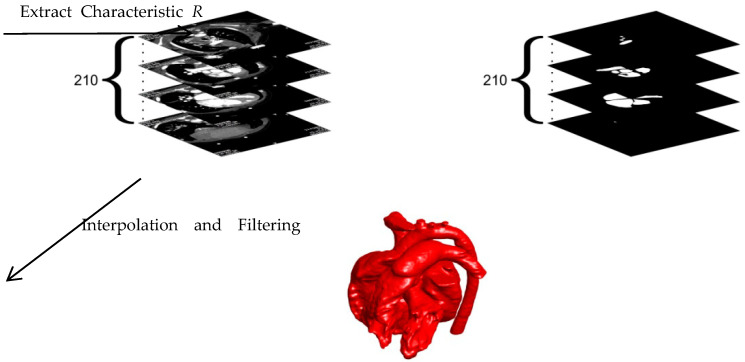
Reconstruction process of the 3D cardiac model.

**Figure 2 entropy-24-00187-f002:**
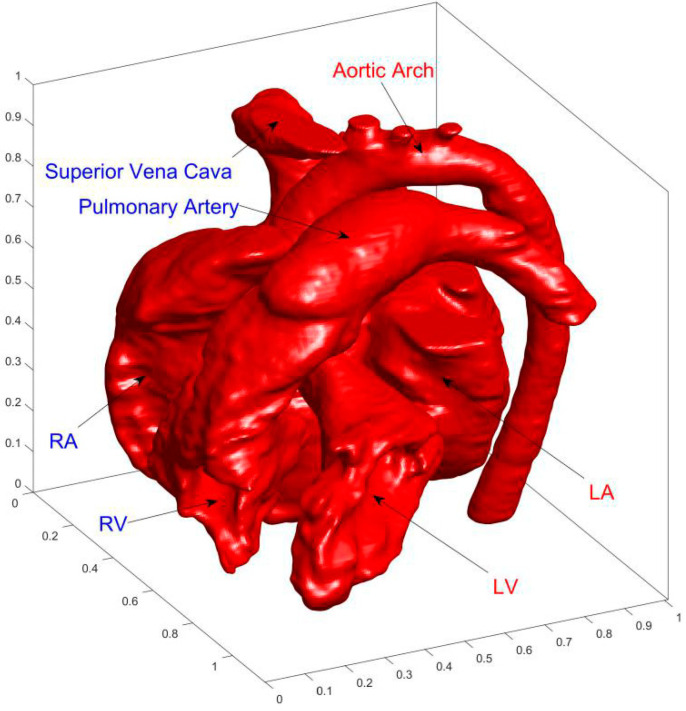
Finial reconstructed3D cardiac model. The acronyms LA, LV, RA, and RV represent left atrium, left ventricle, right atrium, and right ventricle, respectively.

**Figure 3 entropy-24-00187-f003:**
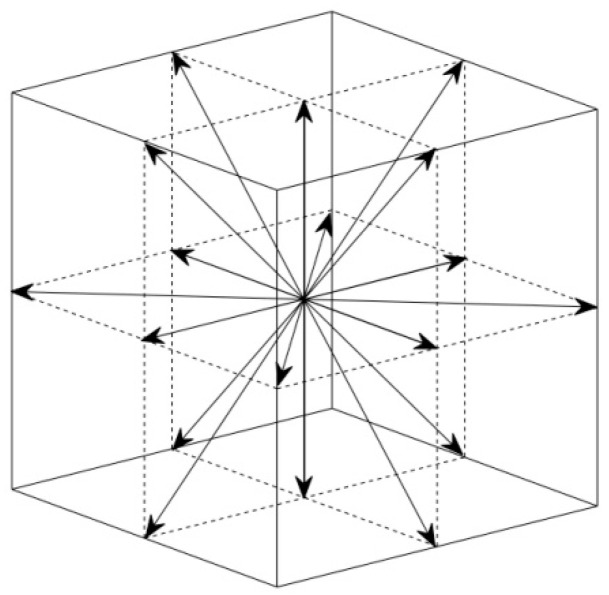
Schematic of the discrete velocities in D3Q19 model. 18 non-zero discrete velocities and a zero discrete velocity.

**Figure 4 entropy-24-00187-f004:**
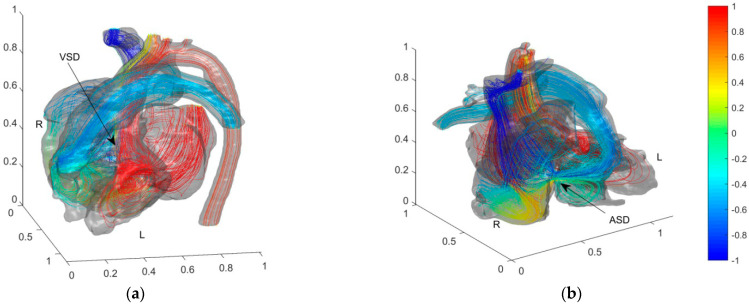
Blood flow and blood oxygen content distribution in the heart of the preoperative child with ASD and VSD. Color of streamlines represents the dimensionless blood oxygen content. Red for relative high oxygen content and blue for relative low blood oxygen content. (**a**) Front view of VSD; (**b**) Front view of ASD.

**Figure 5 entropy-24-00187-f005:**
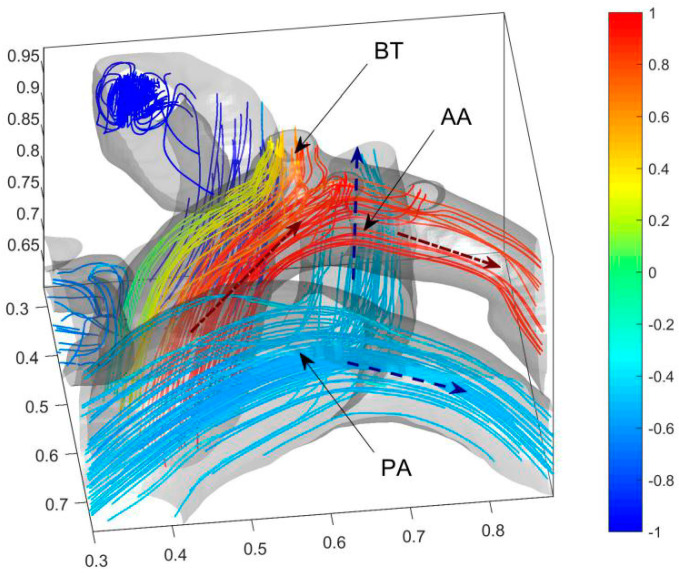
Blood oxygen content distribution of AA and three arterial outlets of the child. Blue dashed arrows (--) represent the directions of the veinal blood and the red dash-dot arrow (-) represents the direction of the arterial blood. The acronym BT, AA, and PA represent the brachiocephalic trunk, aortic arch, and pulmonary artery, respectively.

**Figure 6 entropy-24-00187-f006:**
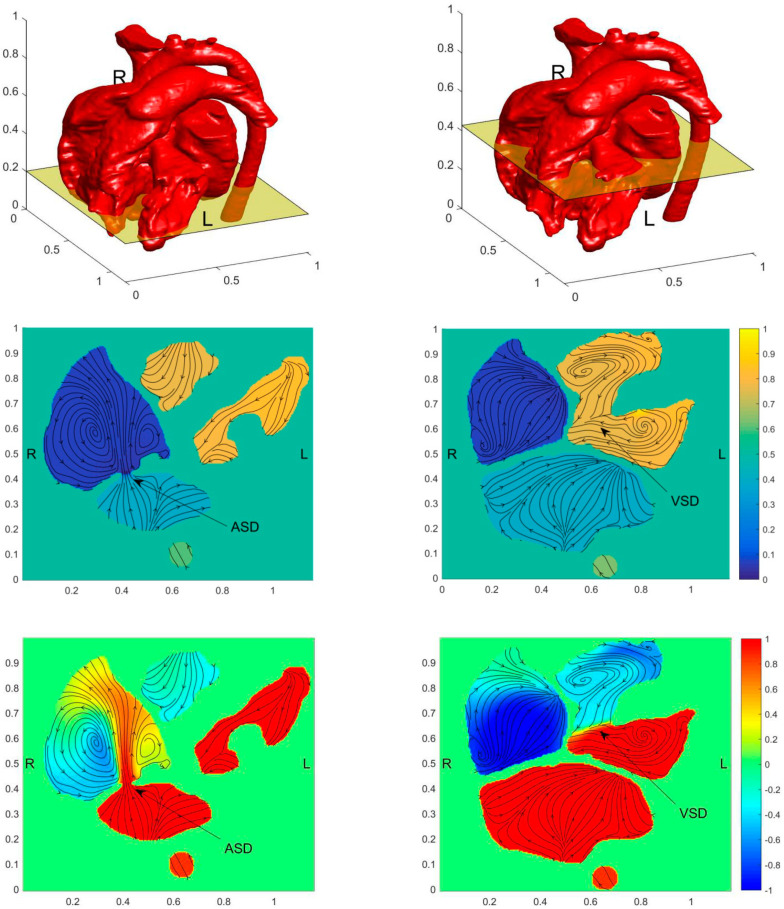
Distribution of pressure field, flow field, blood oxygen content, and entropy production on the horizontal section of ASD and VSD of the preoperative child. The figures in the first row illustrate the positions of the slices. The contours in the second-row show pressure fields, the contours in the third row show the blood oxygen distribution, and the contours in the fourth row show the entropy generation. Figures in the first column have obtained the section at ASD and figures in the second column are obtained at the section at VSD. (**a**) Horizontal section of ASD; (**b**) Horizontal section of VSD.

**Figure 7 entropy-24-00187-f007:**
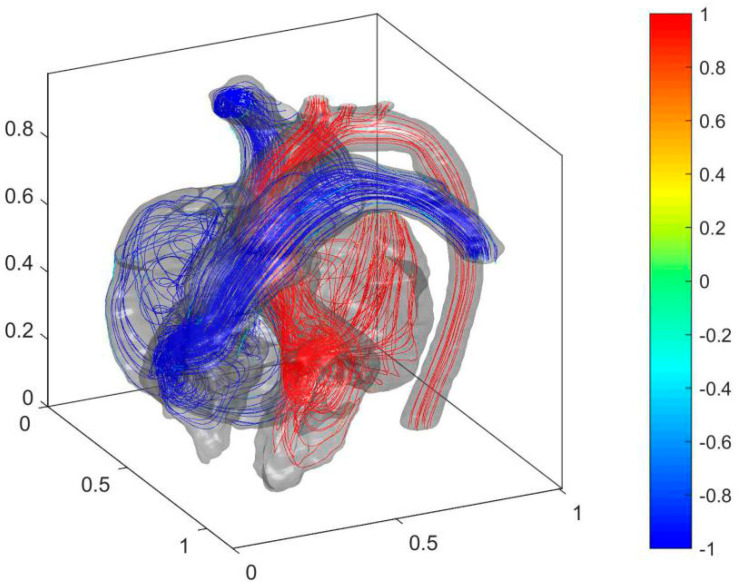
Cardiac flow field and oxygen content distribution in the heart of the postoperative child.

**Figure 8 entropy-24-00187-f008:**
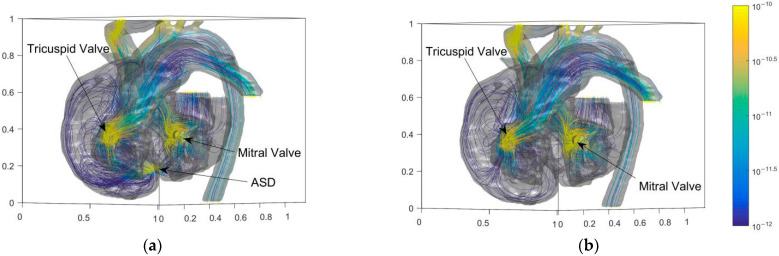
Entropy distributions of blood flow in the heart of the child before and after the operation. (**a**) Before operation; (**b**) After operation.

**Figure 9 entropy-24-00187-f009:**
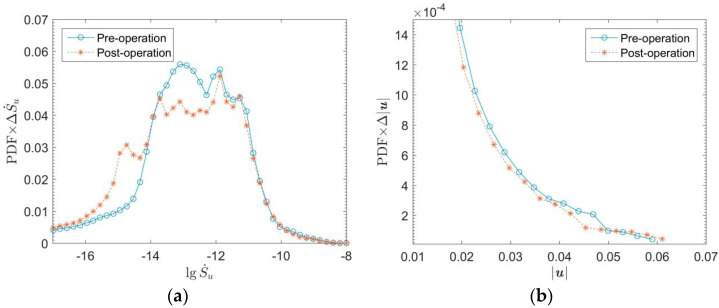
Probability density distributions of entropy production and speed of blood flow in the heart of the child before and after the operation. (**a**) Entropy generation; (**b**) Flow speed.

**Table 1 entropy-24-00187-t001:** Selection of discrete velocities and weight coefficients of D3Q19.

*i*	0	1~6	7~18
|•|	$|c_{i}|=0$	$|c_{i}|=1$	$|c_{i}|=\sqrt{2}$
* **c** _i_ *	(0, 0, 0)	((1, 0, 0)	((1, (1, 0)
		(0, (1, 0)	((1, 0, (1)
		(0, 0, (1)	(0, (1, (1)
*w_i_*	1/3	1/18	1/36

## Data Availability

The data presented in this study are available on request from the corresponding author. The data are not publicly available due to the detailed data is beyond the patient’s consent.

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
