# Peer review of "Simulation of Cardiac Flow under the Septal Defect Based on Lattice Boltzmann Method"

_entropy, 2022, doi:10.3390/e24020187_

Round 1

Reviewer 1 Report

The authors has answered all my questions and I recommend it to be accepted. 

Author Response

Thanks very much

Reviewer 2 Report

Overall, the quality of the paper’s contents has been significantly improved. There has been an improvement in the structure, with ideas better developed and explained. However, before being accepted for publication, the manuscript must be checked for typos, grammar, word redundancies, spaces between words etc.

The following are some points that require minor revisions:

  1. Abstract – leads to lower what?
  2. Introduction – while a detailed explanation of why CFD is necessary is provided, the authors must specify if any other computational studies/methods have been used to simulate ASD and VSD. If yes, include in Introduction. If not, explicitly specify.
  3. Modeling – Line 5 “They came from a child who has both VSD and ASD” redundant with line 1.
  4. Figure 5 – use different colours for the blue and red dashed arrows (maybe black?), since these currently blend in easily with the streamlines.
  5. Figure 6 – consider showing the heart with a plane corresponding to the one used in this analysis; it will help orient the reader.
  6. Since there is no separate discussion section, the authors need to include comparisons of their results with existing results for flow in ASD and VSD patients.
  7. There is only one sentence related to limitations of the model and model assumptions, located at the end of the conclusions. However, given that this paper uses a modelling approach to determine their results, more careful discussion should be given, highlighting how the various limitations will influence the results presented (e.g. lack of fluid-structure interaction).

Author Response

Response: Thanks for the reviewer’s support and valuable suggestions. We have checked the manuscript and improved the words and sentences.

The following are some points that require minor revisions:

  1. Abstract – leads to lower what?
    Response:“It leads to lower efficiency of oxygen binding in the lungs and increases the burden on the heart.” It is corrected in the revised manuscript.
  2. Introduction – while a detailed explanation of why CFD is necessary is provided, the authors must specify if any other computational studies/methods have been used to simulate ASD and VSD. If yes, include in Introduction. If not, explicitly specify.
    Response: Few researches are focused on numerical simulating ASD or VSD due to the complex geometric boundary. It is specified in the revised manuscript.
  3. Modeling – Line 5 “They came from a child who has both VSD and ASD” redundant with line 1.
    Response: The mentioned sentence has been removed in the revised manuscript.
  4. Figure 5 – use different colours for the blue and red dashed arrows (maybe black?), since these currently blend in easily with the streamlines.
    Response: The color of the arrows has been changed into dark blue and dark red. For better , the line width of the arrows is enlarged.
  5. Figure 6 – consider showing the heart with a plane corresponding to the one used in this analysis; it will help orient the reader.
    Response: These two planes have been specified in the revised manuscript.
  6. Since there is no separate discussion section, the authors need to include comparisons of their results with existing results for flow in ASD and VSD patients.
    Response: In the manuscript, we reported the predicted cyanosis and PH caused by VSD. These results are consistent with the clinical report. It has been specified in the revised manuscript.
  7. There is only one sentence related to limitations of the model and model assumptions, located at the end of the conclusions. However, given that this paper uses a modelling approach to determine their results, more careful discussion should be given, highlighting how the various limitations will influence the results presented (e.g. lack of fluid-structure interaction).
    Response: In the last paragraph of conclusion, we gave 2 limitations of present simulation: 1. Moving boundary (FSI is a further topic of moving boundary), 2. Conserved boundary conditions. The possible solutions of these two problems are also specified: IB-LBM and simplified circulation model. The first limitation reckon without the periodic effect of cardiac flow and the second limitations influence the accuracy of blood flow volume and local pressure. The related explain has been added in the revised manuscript.

Round 2

Reviewer 2 Report

Comments have been sufficiently addressed.

This manuscript is a resubmission of an earlier submission. The following is a list of the peer review reports and author responses from that submission.

Round 1

Reviewer 1 Report

The authors did not seem to have considered the pressure difference between LV and RV and LA and RA correctly. Given relatively high pressure in LV compared to RV, it seems very unrealistic to have flow from RV to LV. I recommend assigning pressure boundary conditions for the aorta, pulmonary arteries, veins and pulmonary veins. Also provide more information on the methods section.

Reviewer 2 Report

The research article titled “Simulation of Cardiac Flow Under the Septal Defect Based on lattice Boltzmann method” investigate the change of cardiac flow through lattice Boltzmann method. The method is not novel, but to use it on septal defect is novel. This is certainly a first - or one of a very few - studies addressing cardiac flow under the septal defect before and after the surgery. Thus the significance is high. The research is nice, but the manuscript is not. The author should write it in a more appropriate way.

Abstract

Change “210computed tomography images” 

 to “210 computed tomography images”. Always leave a space between number and words. 

We know for sure that the change of cardiac flow through simulation will change the oxygen content. But what is the possible consequence of changing oxygen content? Rather than correlate the hemodynamics change to the change of oxygen, the author should state the consequence to make the article more impactful and let the reader know the significance of the present research. 

Introduction 

The author should include more literature to highlight the significance of flow change will cause any effect on the septal defect. A good introduction should be 

Paragraph 1: Clinical relevance of the current study 

Paragraph 2: Why CFD is necessary, any previous study done before 

Paragraph 3. What is the challenge and limitation faced by the current field. 

Paragraph 4: Then you propose your solution. 

Please rewrite your introduction.

Suggest to put “We selected a child with both ASD and VSD….after surgery“ from introduction to the method. It seems that the author mixes the method and introduction in the same section, which is quite confusing. 

Method

Please state what algorithm/software was used to reconstruct the geometry from CT images. Please provide details on how many slices on the CT. Please also include whether an ethics approval is secured to do so. 

For the LBM, does the author use in house code or open-source solver? It will be useful if the author can provide the source code or if the author has published the method elsewhere and provide a link so that people can also learn it. 

The free flow boundary conditions seem not justified. I understand that it is hard to measure it, but I suggest that the author provide some literature to support the boundary condition. If not, the author should mention it as a limitation in the discussion. 

Result and discussion

Please specify the high pressure and high oxygen level. e.g. what is the threshold the author uses to define “High”

For figure 2, since it is a streamlined plot, I suggest adding an arrow to let the reader know the direction of flow. 

Again, for that definition of high and low, please give the threshold value. 

From the discussion, I don’t see the effect of post-surgery clearly. Please provide more description on it, e.g. does the post-surgery improve the overall mechanical performance of the heart.

The author mentioned that “It also proves that the energy dissipation of flow decreases after the operation”, I suggest the author continue the unfinished story, so energy dissipation of flow decrease, please provide more evaluation whether dissipation of flow will improve the outcome of surgery or reduce side effect of surgery in the long term.

Reviewer 3 Report

Simulation of Cardiac Flow Under the Septal Defect Based on Lattice Boltzmann Method

In this study, the authors used the Lattice Boltzmann method (LBM) to simulate cardiac flow in a patient with a septal defect. CT images of the heart was used to reconstruct a 3D geometry including the atriums, ventricles, and pulmonary and aortic vessels. It is unclear with the main objective of the study is, besides using the LBM to visualise blood flow in this patient. The authors did not identify any particular clinical question that they were trying to address, state any particular hypothesis. On that note, major revisions should be conducted before this paper is considered for publication.

  1. Clearly state the objective/goal of the study. Provide a clearer introduction and background with development of the rationale/justification of the work. Be clear about what is novel about the work, and what the work adds to our existing knowledge/literature. The relevance of the work must also be highlighted: is it clinical? is it improved theoretical understanding?
  2. A more in-depth evaluation of the existing literature should be presented; there are several existing studies related to computational modelling and CFD in paediatric patients with congenital heart defects. How does this work improve on those existing models? This can be used to support/provide a stronger rationale for the work done in this study.
  3. Clearly delineate between content that should be added as the introduction, which normally includes a background, rationale and objectives; and the methods.
  4. Clearly identify the main metric used in the study, explain why, and how this metric related to the objective (the objective right now is unclear).
  5. Statement related to the use of human subjects data must be included (e.g. IRB approval, Declaration of Helsinki etc.. whatever is used at the author’s institution).
  6. Institute from which the data was obtained should be provided.
  7. Figure captions should include explanation of the acronyms and the main feature of the image that the reader should pay attention to. The variable and units should be indicated on the colour legend. In figure 2, label the anatomy on one image to orient the reader.
  8. Nasal septum?? (section 3)
  9. Actual Inputs to the model are unclear; where did the boundary conditions come from? Are they patient-specific?
  10. I don’t understand the use of entropy production as a metric, but this might relate to an unclear objective.
  11. Abstract needs to be rewritten to better reflect the goals of the study, be less vague about the methods used.
  12. Discussion should highlight what the study added to the literature, and limitations of the work.
  13. Review the entire manuscript for typos, grammar, sentence structure; ensure ideas are properly developed and presented – some points were often confusing to follow; avoid subjective language and vague claims/sentences.